

# Core operational Sentinel-3 marine data product services as part of the Copernicus Space Component

**H. Bonekamp, F. Montagner, V. Santacesaria, C. Nogueira Loddo, S. Wannop, I. Tomazic, A. O'Carroll, E. Kwiatkowska, R. Scharroo, and H. Wilson**

European Organisation for the Exploitation of Meteorological Satellites (EUMETSAT), Darmstadt, Germany

Received: 21 September 2015 – Accepted: 8 November 2015 – Published: 14 January 2016

Correspondence to: H. Bonekamp (hans.bonekamp@eumetsat.int)

Published by Copernicus Publications on behalf of the European Geosciences Union.

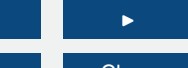
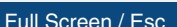


**Abstract**

This paper describes the marine data available from the Marine Centre, part of the Sentinel-3 Payload Data Ground Segment, located at the European Organisation for the Exploitation of Meteorological Satellites (EUMETSAT). The Marine Centre together with the existing EUMETSAT facilities provides a centralised operational service for operational oceanography. These descriptions of the marine data are produced with a focus on a user service perspective. They include the scientific and operational feedback mechanisms on the performance of the services as well as practical information and user support mechanisms.

# 1 Introduction

## 1.1 Copernicus

Copernicus, previously known as GMES (Global Monitoring for Environment and Security), is a European service programme coordinated and managed by the European Commission for the establishment of a European capacity for Earth Observation, see Copernicus (2015). A set of systems collect data from various in situ, airborne, sea-borne and space-borne sensors. These sensor data are processed and provided to the users through a set of data product and modelling services. The services address six thematic areas: land, marine, atmosphere, climate change, emergency management and security, supporting a wide range of applications, including environment protection regional and local planning, fisheries, transport, climate change, sustainable development, civil protection and tourism.

The Copernicus Marine Environment Monitoring Service (CMEMS, see CMEMS (2015)) has been selected by the European Commission to provide the operational oceanography services as part of the Copernicus Programme. These include nowcasting, short term forecasting, and hindcasting of the state of the global

**OSD**

doi:10.5194/os-2015-89

**Core operational Sentinel-3 marine data product services**

H. Bonekamp et al.

ocean and the European regional seas. Similar to those in weather forecasting, computational realistic ocean models together with complex data assimilation systems are run continuously and systematically to provide model and higher level products to downstream, value adding services. These services require a systematic, reliable operational satellite data service which provides data with a high level of availability, timeliness and quality. EUMETSAT is providing in the Copernicus context this marine satellite data service for Sentinel-3, and will provide in years to come also Sentinel-4, 5 and 6 data.

## 1.2 Copernicus Space Component

The provision of the space-borne sensor data within Copernicus is called the Copernicus Space Component (CSC). It is developed under the aegis of the European Space Agency (ESA). First of all, within the CSC, there are both Copernicus dedicated instruments and dedicated satellites. These sensors or satellites are called the Sentinels: Sentinel-1 is a dedicated satellite providing Synthetic Aperture Radar (SAR) imagery for land and ocean services and applications, the first satellite of the series (S1A) was launched on 3 April 2014, with the second one (S1B) due for launch in 2016. Sentinel-2 is also a dedicated satellite providing multispectral high-resolution optical imagery mainly for land services and applications, S2A was launched in June and is still in its commissioning phase.

Sentinel-3, the subject of this paper will provide high-accuracy ocean colour, sea surface temperature and surface topography data. Further details of the Sentinel-3 payload are provided in Sect. 1.3. Sentinel 4 and 5 are instruments dedicated to atmospheric composition and will be flown on the Meteosat Third Generation satellites and Metop Third Generation satellites respectively, see, e.g., Klaes and Holmlund (2014). Finally, the Sentinel-6 or Jason-CS satellite provides high precision radar altimetry data, complementing that of Sentinel-3 as a follow on to the Jason series of satellites, see Scharroo et al. (2015).

**OSD**

doi:10.5194/os-2015-89

**Core operational Sentinel-3 marine data product services**

H. Bonekamp et al.

## 1.3   Sentinel-3 payload

The Sentinel-3 mission, see Donlon et al. (2012) consists of two parts: An optical mission and a surface topography mission. The optical mission is based on two payload instruments. Firstly, the Ocean and Land Colour Imager (OLCI) is a push-broom imag-
ing spectrometer with five cameras, see e.g. Nieke et al. (2012). The joint swath with a total width of 1270 km has a westerly offset against the satellite nadir ground track of approximately 300 km to mitigate for sun glint, see Fig. 1. Each camera has 21 spectral bands in the range of 400–1020 nm, see Table 1. The full resolution sampling is 300 m, reduced is 1 km and the absolute radiometric accuracy requirement is 2 %.
The second optical instrument is the Sea and Land Surface Temperature Radiometer (SLSTR), see Coppo et al. (2013). SLSTR has a near simultaneous nadir and accompanying oblique view. The larger near nadir view swath (1400 km) shares roughly its westerly boundary with that of OLCI (see Fig. 1). The narrower oblique view swath (740 km) is approximately centred over the satellite nadir ground track. SLSTR has
three spectral channels in the Visible (VIS) range (S1, S2, S3), three spectral bands in the Short Wave and InfraRed (SWIR) range (S4, S5, S6) and three in the thermal Infrared (S7, S8, S9) see Table 2. Two additional channels (F1, F2) are also available to detect high temperature events such as gas flares over the ocean.
        The surface topography mission is based on a Synthetic Aperture Radar Altimeter
(SRAL) instrument, see Le Roy et al. (2010). This is a Ku- and C-band nadir-looking radar with Synthetic Aperture Radar (SAR) capabilities to provide sea surface topography measurements in a low (approx. 7 km and a high (approx. 300 m) resolution mode. A dual frequency Microwave Radiometer (MWR), see Bergadà et al., 2010), supports the SRAL to provide the overall sea surface height by providing the wet atmo-
sphere correction. In addition, for the precise determination of the orbit Sentinel-3 provides a Global Navigation Satellite System (GNSS) receiver, a Doppler Orbitography and Radio-positioning Integrated by Satellite (DORIS) instrument, and a laser retro-reflector.

# OSD

doi:10.5194/os-2015-89

**Core operational
Sentinel-3 marine
data product services**

H. Bonekamp et al.



## 1.4 Sentinel-3 ground segment organisation

The Sentinel-3 System includes a Payload Data Ground Segment (PDGS) to perform the satellite data downlink, the data and product processing, dissemination and archiving, and a Flight Operations Segment (FOS) responsible for the spacecraft control. The PDGS is composed of a number of different centres responsible for the near real time and offline processing, dissemination and archiving of the land and respectively marine products, and for the missions performance activities such as instrument and product monitoring, calibration and validation. The processing, dissemination and archiving of the near real time and offline marine products as well as the marine mission performance activities are carried out at the Sentinel-3 Marine Centre located in EUMETSAT, see also Fig. 2, and works in close cooperation and coordination with the ESA led Land Processing and Archiving Centres and Mission Performance Centre, see Sect. 5.

In this paper we explain all aspects of the Sentinel-3 Marine Data Services from an operational user perspective. Section 2 describes the services in the ocean colour, ocean surface topography, and sea surface temperature domains in terms of products and product characteristics. Section 3 explains in detail the EUMETSAT and ESA joint mission performance planning and organisation which will ensure data product quality in terms of overall scientific characterisation as well as in terms of adequacy for operational use by the Copernicus monitoring and other services. Finally, Sect. 4 explains the user interfaces and product dissemination mechanisms of the Sentinel-3 Marine Centre at EUMETSAT.

## 2 Data product services

The definition of the Marine Centre data product services are provided in a dedicated Copernicus Service Level Specification document (SLS, 2015). In the below, the services are explained on the basis of its initial version, 1.0.

Discussion Paper | Discussion Paper | Discussion Paper | Discussion Paper |

**OSD**

doi:10.5194/os-2015-89

**Core operational Sentinel-3 marine data product services**

H. Bonekamp et al.

## 2.1  Generic data product service aspects

The Sentinel-3 data product services are provided with three different timelinesses to address the differing user needs for applications in both the online and offline domains: Near-Real-Time (NRT) products are made available to the users within 3 h after sens-
ing; Short-Time-Critical (STC): products are available to the users within 48 h after sensing, although for several operational oceanography applications (Ocean weather forecasts) this may still be considered as near real time. Non-Time-Critical (NTC) products are available to the users within 1 month after sensing. The standard level 1 are provided globally and the marine level 2 user products are provided for all ocean/water
surfaces depending on an agreed land/sea mask, however, in addition to these products, there are also pre-defined data sets provided for different application areas (e.g. band subsetting) and/or different regions of user interest. A number of different regions have been defined, including several European regions specified by CMEMS. These regions are: the Arctic; the Baltic; the Mediterranean, the Black Sea, the North Atlantic
and the European Seas as a whole, see Fig. 3. Level 0 are not considered as user products but are available to special users, e.g. those users who are supporting the calibration and validation activities for S3.

All Sentinel-3 data products are provided in a Sentinel-specific variation of the Standard Archive Format for Europe (SAFE) format specification. This specification is based
on the concept of eXtensible Markup Language (XML) formatted Data Units (XFDU) called packages or "products", see Fig. 4. The manifest file is in XML format and contains the logical overview of the package together with product metadata. The essential geophysical product (scientific data) are contained in measurement data files, encoded in NetCDF4 format. Quick looks/browse products of the data may also be included as
measurement files. Optional annotation files contain data other than instrument measurement data, e.g. corrections. The information contained in these file can also be common to several measurements data files contained in the same product package.

**OSD**

doi:10.5194/os-2015-89

**Core operational Sentinel-3 marine data product services**

H. Bonekamp et al.

Discussion Paper | Discussion Paper | Discussion Paper | Discussion Paper | Discussion Paper |

More details can be found in the product definition documentation (2013) or in the ESA Sentinel-3 handbook (2013).

## 2.2 Ocean Colour data product service

The Ocean Colour data products service, see Table 3, is based on the OLCI measurements. The OLCI level 1 products, which are used by both the land and marine services, consist of radiometric measurements computed from the instrument digital counts in the 21 bands (see Table 1) and valid at the top of the atmosphere. These measurments are geo-referenced, radiometricly corrected (non-linearity, smear and dark-offset corrections, absolute gain calibration adjusted for gain evolution with time), corrected for stray-light, spatially resampled top of the Atmosphere upweeling radiances specified a ground grid, and annotated with initial pixel classification and auxiliary meteorological data at tie points. The Full Resolution (FR) is approximately 300 m. Data products at reduced resolution (RR = 1200 m), are obtained by averaging the signal of 16 FR pixels: Four adjacent pixels across track by four successive pixel lines along track. Level-1 product processing at FR and RR is the same over the whole globe, land and water surfaces, as well as regional seas (see Fig. 3).

The FR and RR level 2 products (OL_2_WFR and OL_2_WRR see Table 3) consist of parameters in the ocean colour domain derived from the level 1 FR and RR products. Key derived parameters are: water-leaving reflectances in the 16 bands and algal pigment concentrations for open ocean and coastal waters derived using, respectively, the OC4Me (Morel et al., 2007a) and neural network algorithms (Doerffer and Schiller, 2007). Other water parameters are: total suspended matter concentration; diffuse attenuation coefficient; coloured detrital and dissolved organic material absorption. Atmospheric by-products are aerosol optical depth and Angstrom exponent over water. Further products are photosynthetically active radiation over oceans and global coverage integrated water vapour column. The OLCI level 2 products are provided for the global ocean as well as certain regions of interest, as defined in Fig. 3.

Discussion Paper | Discussion Paper | Discussion Paper | Discussion Paper | Discussion Paper |

**OSD**

doi:10.5194/os-2015-89

**Core operational Sentinel-3 marine data product services**

H. Bonekamp et al.

## 2.3 Sea Surface Temperature data product service

The SST data product service is based on the SLSTR measurements as shown in Table 2. The level 1 products consist of calibrated and geolocated radiances and brightness temperatures computed from instrument source packets in the thermal, short wave and visible channels. The SLSTR level 1 products contain: the radiances of the 3 VIS, the 3 SWIR (on the A and B stripe grids), and the 3 MWIR/TIR bands; the Brightness Temperature (BT) for the 3 TIR bands ; and the Brightness Temperature (BT) for the 2 FIR bands. Measurements from the different channels are provided for both the nadir and the oblique view where applicable dependent on the position in the swath. For each channel, the detectors have multiple elements which vary in number according to the channel (see Coppo et al., 2013). These measurements are accompanied with grid and time information, quality flags and error estimates.

The Level 2 products are based on a single Sea Surface Temperature (SST) field derived from the best performing single-coefficient SST in any given part of the swath, plus a number of supporting data fields providing context for the SST fields. The choice of SST is dependent on the view, time of day, and (in planning) dust/aerosol conditions. The measurement data files conform to the GHRSST L2P specification (see The Recommended GHRSST Data Specification GDS). The SST retrieval is based on combinations of brightness temperatures weighted by coefficients which can be defined using modelled radiances followed by regression to an equation whose form accounts for view-geometric and other factors (Sea Surface Temperature (SLSTR) Algorithm Theoretical Basis Document). The SLSTR level 2 products are provided for the global ocean. Datasets are explained in Fig. 3.

## 2.4 Ocean Surface Topography data product service

The Ocean Surface Topography services are mainly based on the SRAL measurements. There are two mutually exclusive measurement modes: A low-resolution (approx. 7 km) measurement mode (LRM) based on pulse limited radar processing, and

Discussion Paper | Discussion Paper | Discussion Paper | Discussion Paper |

**OSD**

doi:10.5194/os-2015-89

**Core operational Sentinel-3 marine data product services**

H. Bonekamp et al.

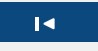
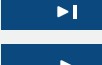
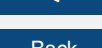
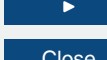
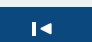
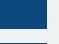
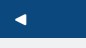
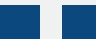
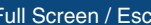

**OSD**

doi:10.5194/os-2015-89

**Core operational Sentinel-3 marine data product services**

H. Bonekamp et al.

a high-resolution (approx. 300 m) SAR measurement mode based on Synthetic Aperture Radar (SAR) techniques. The operational mode is set by an on board geographical mode mask. Currently, the default operational mode is to produce SAR mode measurments over the entire globe (hence high resolution only).

SRAL level 1A products (see Table 4) consist of level 0 unpacked complex radar echoes that have been sorted and calibrated. Also geo-location information is included in this product to allow expert users an easy start towards higher level processing. SRAL Level 1B-S (S stands for stack) containing geo-located, calibrated azimuth formed complex (I and Q) after slant/Doppler range correction over a fixed point on the ground-track. The echoes from the SRAL level 1A products are used. The level 1B products consist of the 20 Hz averaged measurements (also for the LRM mode).

SRAL level 2 products are based on processing of the SRAL level 1B products. The key SRAL level 2 physical quantities derived are surface height SSH, range, the normalized backscatter, sea ice freeboard, siginificant wave height, 10 m wind speed. The accuracy of the Surface height measurement is modulated by the NRT, STC and NTC timeliness as the restituted, the preliminary, and the final precise satellites orbits are calculated with design orbit accuracies of resp 10, 4 and 3 cm (RMS). At level 2, the measurments of SRAL and MWR are combined and annotated similar to what is known from other altimeter missions (for example Jason-2). Three measurements files are generated: a "standard" data file, containing the standard 1 and 20 Hz Ku and C bands parameters; a "reduced" measurement data file, containing a subset of the main 1 Hz Ku band parameters; and a "enhanced" data file, containing the standard 1 and 20 Hz Ku and C bands parameters, the waveforms and the associated parameters necessary to reprocess the data.

## 2.5   Water Quality Monitoring data set service

In addition to the three instrument-based services from Sects. 2.2 to 2.4, there is also the Water Quality Monitoring data set service provided only in NRT. This service consists of collecting the OLCI FR and RR level 2 water-leaving reflectance data files for

the 16 bands used in Chlorophyll concentration retrievals (see sub Sect. 2.2) and the brightness temperatures measurement data files for the SLSTR infrared bands (S7, S8, S9) as calculated in the SLSTR level 1 production (see Sect. 2.3). This service is only available for the European Seas region (see Fig. 3) and the data sets are only available from the Online Data Archive (ODA) and the Data Centre (DC) (see Sect. 4).

## 3 Mission performance

### 3.1 Mission performance framework

Ensuring the Sentinel-3 Mission performance in terms of an operational service with the best quality data for the users is a joint activity of ESA and EUMETSAT. An ESA/EUMETSAT joint Cal/Val plan (Rebhan et al., 2014b) is maintained to link the measurement uncertainties with the individual calibration and validation tasks performed by the various entities. As depicted in Fig. 5, a framework of entities has been set-up to deal with anomaly detection and investigation, online and offline instrument and product monitoring, calibration and validation, and product evolutions. The Mission Performance Framework is a joint ESA-EUMETSAT construct operating according to mutually agreed rules. The activities are overseen by the joint mission management. Major changes in the data products services are only taken after endorsement by the Copernicus programme.

### 3.2 Quality working groups

The Sentinel-3 Quality Working Groups (QWG) are advisory groups which support ESA and EUMETSAT on Sentinel-3 data quality aspects. The QWGs bring key users, scientists, and project engineers together regularly to consider the results of relevant Mission Performance Framework activities and to provide recommendations to the ESA and EUMETSAT mission management, to ensure the required level of data quality is maintained throughout the mission lifetime and to contribute to improvements to the

**OSD**

doi:10.5194/os-2015-89

**Core operational Sentinel-3 marine data product services**

H. Bonekamp et al.

Discussion Paper | Discussion Paper | Discussion Paper | Discussion Paper |

data quality taking into account their actual operational use. The QWGs also deal with matters related to the evolution of requirements and data products (e.g. re-processing campaign recommendations, algorithm evolutions, etc.). Three QWGs are foreseen for the data product services as described in Sects. 2.2 (OLCI), 2.3 (SLSTR) and 2.4

(SRAL/MWR). The QWG are joint entities covering both the S3 marine and land services, although the latter are not further explained in this paper. Finally, an additional QWG covering the Precise Orbit Determination aspects is in place covering the orbit determination activities for the Sentinel 1, 2 and 3missions. The POD QWG is closely linked with SRAL/MWR QWG.

### 3.3 Mission performance activities

The ESA led mission performance activities will be performed via the Mission Performance Centre (MPC) service contract, which deals with all the land related aspects of the Sentinel-3 mission performance, see Bruniquel et al. (2015), and provides support to the EUMETSAT marine mission performance activities. Dedicated, instrument

focussed, Expert Support Laboratories (ESLs), address the instrument performance and the various Sentinel-3 calibration and validation tasks.

The ESA facilities are complemented by Marine Mission Performance Monitoring Facility and the in-house experts at the Marine Centre in EUMETSAT.

### 3.4 Sentinel-3 Validation Team

The mission performance activities will be complemented by the activities of the Sentinel-3 Validation Team (S3VT) who are external experts or users performing activities which support the Sentinel-3 calibration and validation activities. The S3VT consists of 4 subgroups which are co-chaired by the ESA and EUMETSAT domain experts: Sea Surface Temperature, Ocean Colour, Altimetry and Land Applications. Users can

join the S3VT via a rolling call for an Announcement of Opportunity, once their proposal for validation activities is accepted, see S3VT website.

**OSD**

doi:10.5194/os-2015-89

**Core operational Sentinel-3 marine data product services**

H. Bonekamp et al.

### 3.5 EUMETSAT in-house mission performance activities

EUMETSAT's mission performance activities are guided by the joint cal/val plan (Rebhan et al., 2014b) and are carried out in collaboration with ESA at L1 and autonomously for L2 marine aspects. These activities rely on in house measurements, product and operations experts in close interaction with system engineering knowledge of the various components of the PDGS. A multi-mission approach is taken where synergies exist with similar activities within the other EUMETSAT programmes (see e.g., Klaes and Holmlund, 2014).

## 4 Product dissemination and user support

The right hand side of Fig. 2 depicts the product dissemination facilities of the Sentinel-3 Marine Centre. In addition, EUMETSAT will also provide a series of web-based services, which have been developed to support users in the access and exploitation of the Marine data.

### 4.1 EUMETCAST

EUMETCAST is a multi-service dissemination system based on standard Digital Video Broadcast (DVB) technology. It uses commercial geostationary telecommunication satellites to multi-cast files (data and products) to a wide user community. A One-stop-shop delivery mechanism allows users to receive many data streams (not only Sentinel-3) via one low cost reception station. EUMETCAST services cover Europe, Africa and South America, see EUMETSAT website for details of individual data offers in these regions.

Discussion Paper | Discussion Paper | Discussion Paper | Discussion Paper | Discussion Paper |

**OSD**

doi:10.5194/os-2015-89

**Core operational Sentinel-3 marine data product services**

H. Bonekamp et al.

## 4.2  The Sentinel-3 Online Data Access

The Sentinel-3 Online Data Archive (ODA) is a mission dedicated online rolling archive containing 1 month of products supporting ftp/http access over the Internet. All data products and datasets as described in Sect. 2 are available from the ODA.

## 4.3  The EUMETSAT Data Centre

The EUMETSAT Data Centre is multi mission facility providing the long term storage of the complete historical coverage of all EUMETSAT's missions and will also include the data from the Sentinel missions operated by EUMETSAT (S3/4/5/6). Users can browse, order, and retrieve data from EUMETSAT's extensive catalogue of products. All data products and datasets as described in Sect. 2 are available from the DC.

## 4.4  Catalogue, registration and user support

The Sentinel-3 user products will be included in the online catalogue for all the EUMET-SAT data and product services. This catalogue is called the Product Navigator (PN), http://navigator.eumetsat.int/. The PN includes simple, thematic as well more complex, extended search capabilities allowing a spectrum of users ranging from novice, new and interested, to experienced, and operational to find what they need. The collection entries are compatible with ISO 19115/19139 standards and conform to the EU INSPIRE directive. All PN entries include a product description and elementary information such as coverage; dissemination; file naming formats and the links to access the product.

To access Sentinel-3 data users will be asked to first register via the EUMETSAT Earth Observation Portal (EOP). Once an account has been created, users can log in to view and modify their profile, service subscriptions and licence arrangements although these are not needed for Sentinel-3. The EOP provides a single entry point to Sentinel-3 data whether disseminated via EUMETCAST, downloaded via the Online

**OSD**

doi:10.5194/os-2015-89

**Core operational Sentinel-3 marine data product services**

H. Bonekamp et al.

Data Access Service or ordered through the Long-term Archive, the Data Centre. Via the EOP users can also subscribe to the User Notification Service (UNS) The UNS provides information on the status of the satellites, derived products and data access services. The UNS gives up-to-date information on scheduled maintenance outages, new product releases and enhancements and service alerts when anomalies occur. The system is used for the EUMETSAT Meteosat, Metop, and Jason satellite data products services (see, e.g., Klaes and Holmlund, 2014) well as those of EUMETSAT third-party services. Sentinel-3 UNS information will be introduced at the start of the operational phase.

*Acknowledgements.* The authors thank their colleagues from the wider EUMETSAT Sentinel-3 team and their ESA counterparts for their collaboration and support.

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



**Table 1.** OLCI spectral bands. Name and central wavelengths (bandwidth) in nanometers, see Nieke et al. (2012).

| Name | Wavelength | Name | Wavelength | Name | Wavelength |
|------|-----------|------|-----------|------|-----------|
| Oa01 | 400 (15) | Oa08 | 665 (10) | Oa15 | 767.5 (2.5) |
| Oa02 | 412.5 (10) | Oa09 | 673.75 (7.5) | Oa16 | 778.75 (15) |
| Oa03 | 442.5 (10) | Oa10 | 681.25 (7.5) | Oa17 | 865 (20) |
| Oa04 | 490 (10) | Oa11 | 708.75 (10) | Oa18 | 885 (10) |
| Oa05 | 510 (10) | Oa12 | 753.75 (7.5) | Oa19 | 900 (10) |
| Oa06 | 560 (10) | Oa13 | 761.25 (2.5) | Oa20 | 940 (20) |
| Oa07 | 620 (10) | Oa14 | 764.375 (3.75) | Oa21 | 1020 (40) |

**OSD**

doi:10.5194/os-2015-89

**Core operational Sentinel-3 marine data product services**

H. Bonekamp et al.

**Table 2.** SLSTR spectral bands. Name and central wavelengths (bandwidth) in nanometers (Coppo et al., 2013). (S1, S2, S3), (S4, S5, S6) and (S7, S8, S9) are respectively, Visible (VIS), Short Wave and InfraRed (SWIR) and Infrared bands. (F1, F2) are Fire detection bands.

| Name | Wavelength | Name | Wavelength | Name | Wavelength | Name | Wavelength |
|------|------------|------|------------|------|------------|------|------------|
| S1 | 555 (20) | S4 | 1375 (15) | S7 | 3740 (380) | F1 | 3740 (380) |
| S2 | 659 (20) | S5 | 1610 (60) | S8 | 10850 (900) | F2 | 10850 (900) |
| S3 | 865 (20) | S6 | 2250 (50) | S9 | 12000 (1000) | | |

**OSD**

doi:10.5194/os-2015-89

**Core operational Sentinel-3 marine data product services**

H. Bonekamp et al.

**Table 3.** Ocean Colour user data products (OLCI). All products are available from the monthly online rolling archive (ODA) and the long-term archive (DC), see Sect. 3. EUMETCAST dissemination is indicated by (E). Granularity: the products are provided as either 3 min Product Data Units (PDU's) or Daylight orbits. The sizes are given for a full orbit and are an approximation based on compression assumptions.

| ID | Level | Resolution | NRT | STC | NTC | Size (Gb) |
|---|---|---|---|---|---|---|
| OL_1_EFR | 1 | Full | PDU (E) | – | PDU | 21.5 |
| OL_1_ERR | 1 | Reduced | Daylight orbit | – | Daylight orbit | 1.4 |
| OL_2_WFR | 2 | Full | PDU | – | PDU | 14.2 |
| OL_2_WRR | 2 | Reduced | Daylight orbit (E) | – | Daylight orbit | 0.95 |

**OSD**

doi:10.5194/os-2015-89

**Core operational Sentinel-3 marine data product services**

H. Bonekamp et al.

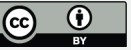

**Table 4.** Sea Surface Temperature user data products (SLSTR). All products are available from the monthly online rolling archive (ODA) and the long-term archive (DC), see Sect. 3. EUMETCAST dissemination is indicated by (E). Granularity: the products are provided as 3 min Product Data Units (PDU's). The sizes are given for a full orbit and are an approximation based on compression assumptions.

| ID | Level | Resolution | NRT | STC | NTC | Size (Gb) |
|---|---|---|---|---|---|---|
| SL_L1_RBT | 1 | Full | PDU | – | Full Orbit (South Pole–South Pole) | 29.0 |
| SL_L2_WST | 2 | Full | PDU(E) | – | Full Orbit (South Pole–South Pole) | 0.75 |

**OSD**

doi:10.5194/os-2015-89

**Core operational Sentinel-3 marine data product services**

H. Bonekamp et al.

**Table 5.** Ocean Surface Topography data products (SRAL). All products are available from the monthly online rolling archive (ODA) and the long-term archive (DC), see Sect. 3. EUMETCAST dissemination is indicated by (E). The sizes are given for a full orbit and are an approximation based on compression assumptions. The SR_1_A and SR_1_BS products are in planning (see Rebhan et al., 2014a) and the product sizes are a rough estimation.

| ID | Level | Resolution | NRT | STC | NTC | Size (Gb) |
|---|---|---|---|---|---|---|
| SR_1_A | 1a | Full | – | Half orbit | Half orbit | 17 |
| SR_1_BS | 1bs | Full | – | Half orbit | Half orbit | 17 |
| SR_1_SRA | 1b | Full | Full orbit (E) | Half orbit | Half orbit | 0.4 |
| SR_2_WAT | 2 | Full | Full orbit (E) | Half orbit (E) | Half orbit | 0.2 |

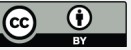

**OSD**

doi:10.5194/os-2015-89

**Core operational Sentinel-3 marine data product services**

H. Bonekamp et al.

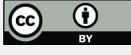

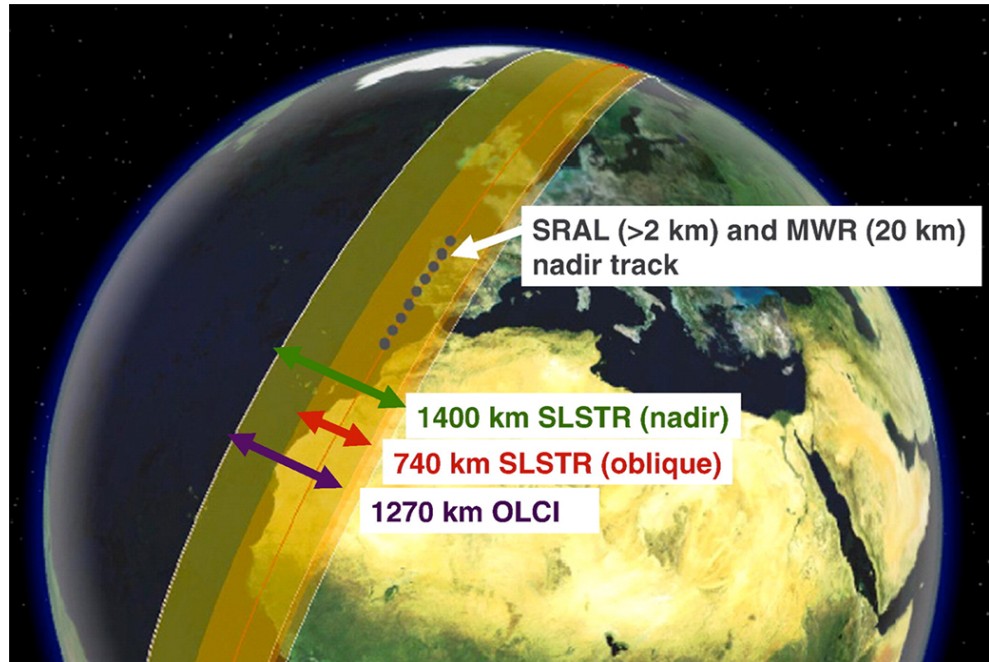

**Figure 1.** Schematic overview of the Sentinel-3 instrument swaths (Courtesy ESA).

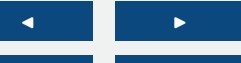

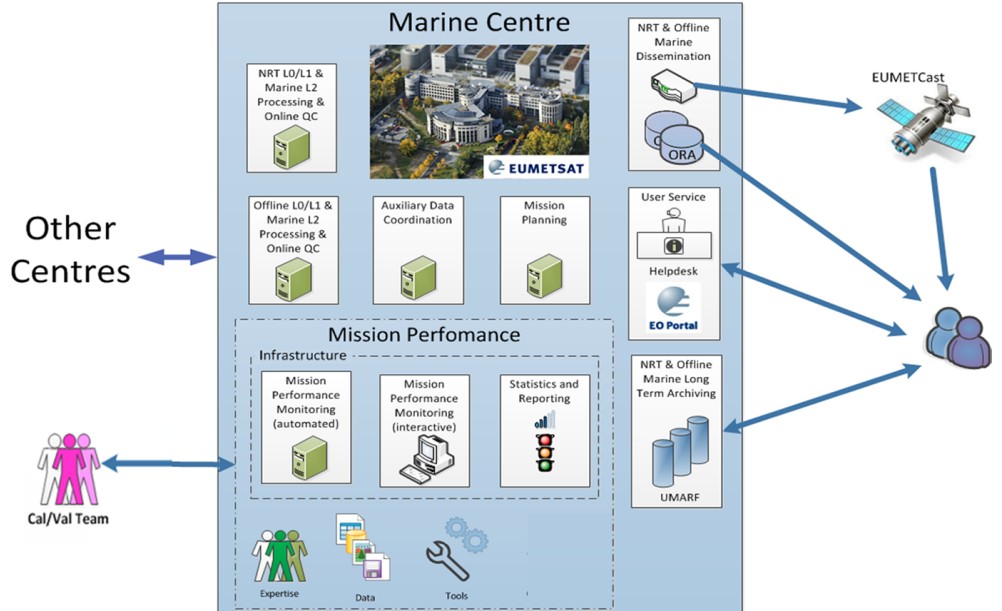

**Figure 2.** Schematic overview of the Sentinel-3 Marine Centre at EUMETSAT. The Marine Centre consists of mission planning, data processing, mission performance, data dissemination and user support components.

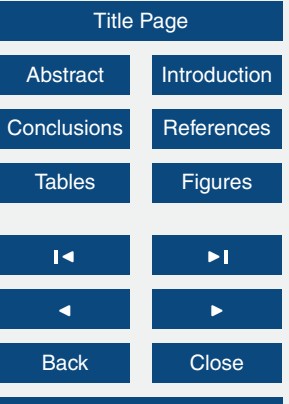

OSD

doi:10.5194/os-2015-89

**Core operational Sentinel-3 marine data product services**

H. Bonekamp et al.

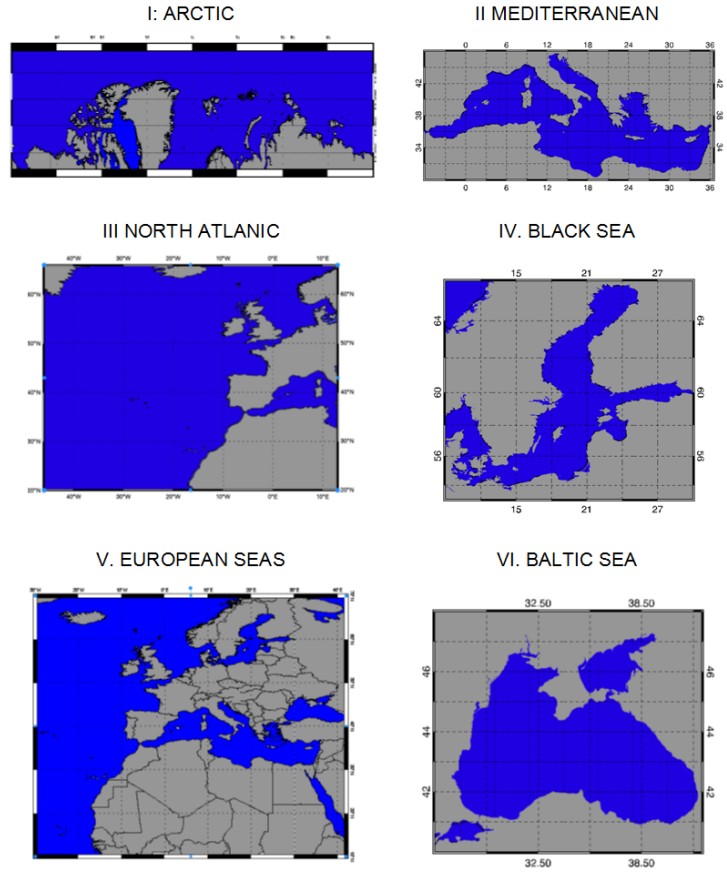

**Figure 3.** Maps of the Regional Data Sets. The longitude (latitude) ranges are respectively for (i) Arctic seas, 180° W–180° E (66–90° N); (ii) Mediterranean Sea, 6° W–36.5° E (30–46° N); (iii) North-Atlantic, 46° W–13° E (20–66° N); (iv) Baltic Sea, 9.25–6.5° E (53–66.85° N); (v) European seas, 30° W–42° E (10–70° N); (vi) Black Sea, 26.5–40.0° E (40–48° N).

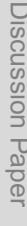

# OSD

doi:10.5194/os-2015-89

**Core operational Sentinel-3 marine data product services**

H. Bonekamp et al.

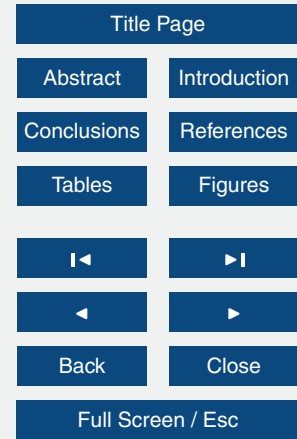

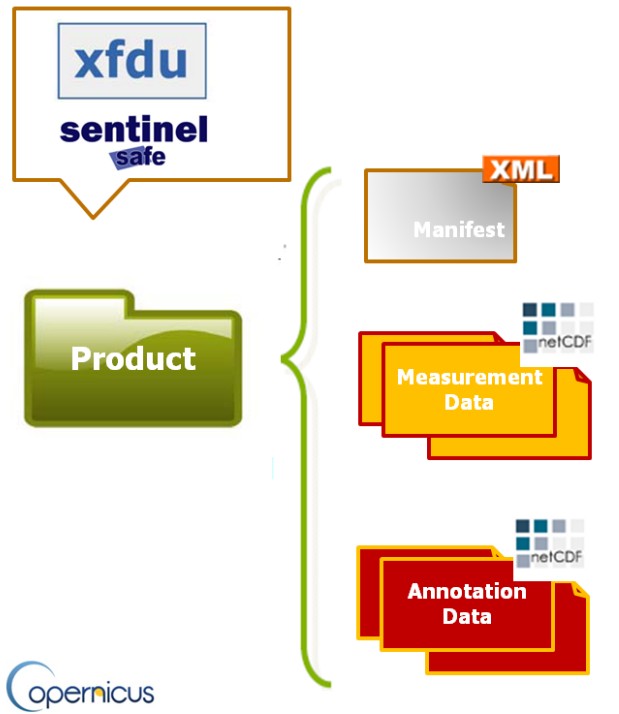

**Figure 4.** Schematic overview of the Safe format.

Discussion Paper | Discussion Paper | Discussion Paper | Discussion Paper | Discussion Paper |

**OSD**

doi:10.5194/os-2015-89

**Core operational Sentinel-3 marine data product services**

H. Bonekamp et al.

**OSD**

doi:10.5194/os-2015-89

**Core operational Sentinel-3 marine data product services**

H. Bonekamp et al.

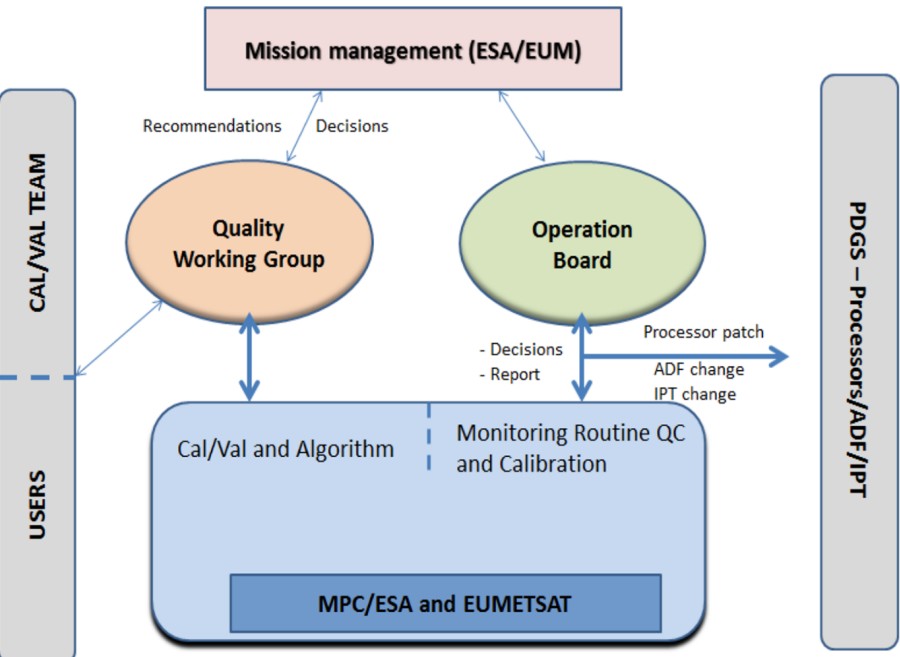

**Figure 5.** Schematic overview of the joint ESA and EUMETSAT Sentinel-3 Mission Performance Framework. An operational board deals with the daily operations and the resolution of anomalies. On the left side (in grey) there are the users and the user-based Sentinel-3 Validation Team (see Sect. 3.4) interfacing with the Quality Working Groups, see Sect. 3.2. At the bottom are the core mission performance activities of the MPC (see Sect. 3.3) and of the EUMETSAT Marine Centre (Sect. 3.5). Changes in data products services are generated by updates and patches of the data processors of the Instrument Parameters Tables (IPTs) and of the auxiliary data files (ADFs) in the PDGS.