# Peer review of "Core Operational Sentinel-3 Marine Data Product Services as part of the"

_Ocean Science, 2015_

## Referee Comment (RC1) · L. A. Breivik (Referee) · 20 Feb 2016

The paper does not present major scientific findings, but gives the reader and potential user of Sentinel-3 ocean data a comprehensive overview of the mission, and its data and products. It will serve as a useful reference for the Sentinel-3 ocean mission.

The paper is well structured, sufficient length (concise enough to avoid being boring)

It is absolute the right timing for publication, a few days after launch of the satellite. Hopefully will the nice data and products described in the paper be available for users within not too far time. And that brings me to my first suggestion: A few words on the time line. When can we expect the data and service to the users ?

[Figure]

Some small comments:

1.1. Copernicus, second paragraph: " to provide model ....value added services, CMEMS require a systematic, reliable ... service " (in order to be sure which service we speak about)

2.1 Generic data products ... First paragraph: about the different timeliness. This is all fine, however there has the last year in various fora been a lot of talk on faster data access and the QRT (Quasi-Real-Time) concept has been introduced, referring to 20 minutes- 1 hour. This is in line with modern demands for nowcasting purposes. Is the mission flexible enough to potentially adjust to these new requirements ? This must have been discussed internally in EUMETSAT, and a few words on the topic might be good and increase the relevance.

About level-1 global and level-2 with land-mask: Is it correctly understood that the S-3 level-2 data from EUMETSAT comes over the ocean, and only over the ocean, while level 1 data will be provided globally (no land mask). This is important e.g. for users who are interested in radiances/brightness temperatures in general. If correct understood make this even more clear!

2.2 Ocean Colour..., second paragraph: "as certain regions of interest" -> "as the Regional Seas" Fig 3 map of Regional Seas, by the way, has mixed the Black and the Baltic in the figure text.

Finally: about Section 4, Product dissemination, what is the relation between the EUMETSAT dissemination facilities and the various "Sentinel Data Hubs" , collaborative, scientific etc, which the user might have heard about and expect to also get Sentinel 3 data from ? A few words on this !
* * *

---

## Referee Comment (RC2) · Anonymous Referee #2 · 10 Mar 2016

The paper by Bonekamp and coauthors presents the operational product services of the Sentinel-3 mission for marine applications. As a such, it is not a research paper with original contributions. Nevertheless, it is relevant since we do expect that Sentinel-3 will significantly contribute to improve our understanding of the ocean. In my opinion, the paper is timely (Sentinel-3 was launched a few weeks ago) and clear. Besides, it may be a good entrance paper to Sentinel-3 data, from which it can be reached the key literature related to the algorithms used. Consequently, I recommend its publication if it fits the policy of OS

I only have some minor comments:

1. It was not evident to me that '(Sea Surface Temperature (SLSTR) Algorithm Theo-

retical Basis Document)' in lines 21-22 was a reference until I scanned the Reference
list. Is there a clearer way of writing it? e.g '(see the details in the ATBD in the reference
list)' or something similar?

2. Figure 3 I suggest to create a single map with boxes showing the different areas and
write a table with their precise limits.

———————————————————

---

## Author Comment (AC1) · 5 Apr 2016

Dear Referee 1 (dear Lars Anders),

> The paper does not present major scientific findings, but gives the reader and potential user of Sentinel-3 ocean data a comprehensive overview of the mission, and its data and products. It will serve as a useful reference for the Sentinel-3 ocean mission. The paper is well structured, sufficient length (concise enough to avoid being boring) It is absolute the right timing for publication, a few days after launch of the satellite. Hopefully will the nice data and products described in the paper be available for users within not too far time. And that brings me to my first suggestion: A few words on the time line. When can we expect the data and service to the users ?

[Figure]

Answer: Thank you for your review of the paper. We agree that this paper is not presenting major scientific findings. Its main merit is to provide for this OS special issue a synoptic overview of the Sentinel-3 Marine Centre at EUMETSAT as the core capability to provide operational services of Sentinel-3 satellite data up to level 2 to the operational user community and in particular to CMEMS. Sentinel-3 was indeed successfully launched on 16 February 2016 and the instrument and production chains are currently in commissioning. To answer your question about availability of the data: It is not possible to give specific dates in this moment in time. Decisions on a pre-operational or operational status of the services follow formal commissioning reviews conducted together with ESA. Announcement will be made at the dedicated EUMETSAT and ESA web pages. However, the rough planning is that first product data releases will be available to the Sentinel-3 validations teams early summer 2016 and to all users end the end of 2016. The schedule of the individual mission may vary.

Related to the small review comments:

> 1.1. Copernicus, second paragraph: " to provide model ....value added services, CMEMS require a systematic, reliable ... service " (in order to be sure which service we speak about)

Answer: In section 1.1: " The services" will be replaced by "The CMEMS services"

> 2.1 Generic data products ... First paragraph: about the different timeliness. This is all fine, however there has the last year in various fora been a lot of talk on faster data access and the QRT (Quasi-Real-Time) concept has been introduced, referring to 20 minutes- 1 hour. This is in line with modern demands for nowcasting purposes. Is the mission flexible enough to potentially adjust to these new requirements ? This must have been discussed internally in EUMETSAT, and a few words on the topic might be good and increase the relevance.

Answer: This is an interesting question, however hard to answer at this moment in time and an answer is beyond the objective of this paper (and perhaps also beyond the

objectives of the Marine Centre). The current targets are to commission the agreed processing baseline and related services into a fully operational status. Sentinel-3 is a Copernicus user driven mission. User requests for substantially improved services have to be addressed to the European Commission. Accordingly, at the end of this paragraph the sentence "Major future updates of the services to the users are managed by the European Commission. " has been added.

> About level-1 global and level-2 with land-mask: Is it correctly understood that the S-3 level-2 data from EUMETSAT comes over the ocean, and only over the ocean, while level 1 data will be provided globally (no land mask). This is important e.g. for users who are interested in radiances/brightness temperatures in general. If correct understood make this even more clear!

Answer: The following sentence has been added to the respective paragraph: "The marine level 2 services roughly extend 30 km land inwards and include major lakes."

> 2.2 Ocean Colour..., second paragraph: "as certain regions of interest" -> "as the Regional Seas" Fig 3 map of Regional Seas, by the way, has mixed the Black and the Baltic in the figure text.

Answer: The last sentence of this paragraph has been adjusted accordingly. Figure 3 has been modified (also in response to comments of the referee 2) and the mix-up of the Baltic and the Black Sea will be corrected.

> Finally: about Section 4, Product dissemination, what is the relation between the EU-METSATdissemination facilities and the various "Sentinel Data Hubs" , collaborative, scientific etc, which the user might have heard about and expect to also get Sentinel 3 data from ? A few words on this

Answer: We agree that this is an interesting question from the user perspective. However all sentinel-3 data and user services are not easily explained. In this paper we have to confine ourselves describing only the services for the marine operational com-

munity. Hence, we describe only the data and user services as referenced in SLS, 2016 (Service level Specification). Note that this reference has updated in due time. In the updated version of the paper we make this explicitly clear to the reader by the additional sentence: "It has to be understood that the described data access and user support services strictly follow the related EUMETSAT Copernicus Operational Service Level Specification (SLS, 2016). Other Sentinel-3 data and user services, as for example for the Sentinel-3 level 2 land products, are not covered in this paper."

---

## Author Comment (AC2) · 5 Apr 2016

**H. Bonekamp et al.**

hans.bonekamp@eumetsat.int

Dear Referee 2 ( anonymous),

General Answer: The authors thank the referee for the constructive review. It is agreed that the paper is not a research paper with scientific findings. Its main merit is to provide for this OS special issue a synoptic overview of the Sentinel-3 Marine Centre at EUMETSAT as the core capability to provide operational services of Sentinel-3 satellite data up to level 2 to the operational user community and in particular to CMEMS.

> 1. It was not evident to me that '(Sea Surface Temperature (SLSTR) Algorithm Theoretical Basis Document)' in lines 21-22 was a reference until I scanned the Reference
list. Is there a clearer way of writing it? e.g '(see the details in the ATBD in the reference list)' or something similar?

Answer to 1.: the authors acknowledge that the reference could be more clearly written. The text is adjusted as follows: "The SST retrieval is based on combinations of brightness temperatures weighted by coefficients which can be defined using modelled radiances followed by regression to an equation whose form accounts for view-geometric and other factors. The precise algorithms are described in the Sea Surface Temperature (SLSTR) Algorithm Theoretical Basis Document (SLSTR-ATBD, 2012)."

> 2. Figure 3 I suggest to create a single map with boxes showing the different areas and write a table with their precise limits.

Answer to 2.: The authors agree with the referee that a single map would improve the indication of the agreed regional data sets. A new map has been created and is attached to this reply. The letters indicate the data sets. Figure 3 and its caption will be updated accordingly.

[Figure]

[Figure]

**Fig. 1.** new map for figure 3